

# Surficial sediment texture database for SW Iberia Atlantic Margin

Susana Costas, Margarida Ramires, Luisa B. de Sousa, Isabel Mendes, Oscar Ferreira

Centre for Marine and Environmental Research-CIMA, University of Algarve, Faro, 8005-139, Portugal

*Correspondence to*: Susana Costas (scotero@ualg.pt)

5 **Abstract.** Assessing the impact of changes on the environment driven by natural or anthropogenic forcers includes the comparison between antecedent and post-event conditions. The latter is particularly relevant in order to better understand to which extent those changes actually impact or alter a particular environment and associated services and to determine the resilience of a system. In this regard, it turns essential to create or provide databases to inform about baseline conditions. Here, we present a database that integrates surficial sediment samples collected and analysed for textural characterization within the 10 frame of a series of research projects over circa 20 years. Collected samples along the southwestern Atlantic Margin of the Iberian Peninsula extend from estuaries and beaches to the adjacent continental shelf. For the case of the more dynamic environments, namely coastal sandy barriers, samples were repeated over time in order to capture the intrinsic variability of the system. Examples of the utility of this dataset for a variety of purposes and environments are also included within this manuscript through three examples. Therefore, here we show the added value of the database as it can be used to assess the 15 impact of a particular event or activity at an estuary by providing baseline conditions, evaluate the continental shelf sediment suitability for nourishment activities, or to contribute to the morphodynamics understanding and classification of beaches. Finally, it is worth stating the importance of such databases to analyse medium to long-term variability as the one induced by sea level rise, changes in storminess or by human activities. The data presented here are in open access at https://doi.pangaea.de/10.1594/PANGAEA.883104.

## 1 Introduction

Grain size is the most fundamental physical property of sediments. It has been found to determine to a great extend the mode, distance and amount of material that can be transported by a fluid (e.g. Bagnold, 1941;Bagnold, 1956;Rijn, 1993;Shields, 1936;Soulsby, 1997); the slope of beaches (e.g. Bascom, 1951;Sunamura, 1984); the permeability and stability of structures in civil engineering (e.g. Look, 2007;Terzaghi et al., 1996); kinetic reactions and affinities of heavy metals and other 25 contaminants (e.g. Ackermann et al., 1983;Andrieux-Loyer and Aminot, 2001;Krumgalz, 1989); or even the distribution of benthic communities (e.g. Cahoon et al., 1999;Gibson and Robb, 1992;McLachlan, 1996;Yang et al., 2008). Likewise, and assuming the aforementioned controls that this property may exert, grain size is used as environmental proxy to identify and reconstruct sediment transport pathways (e.g. Gao and Collins, 1992;Gao and Collins, 1994;Le Roux, 1994;Le Roux and Rojas, 2007), past records of wind intensity (Lindhorst and Betzler, 2016), changes in the processes generating windblown



sediments (Vandenberghe, 2013), changes in the strength of continental shelf currents and precipitation regimes over time (Gyllencreutz et al., 2010), or to infer driver-related energy gradients (Rosa et al., 2013).

The above also suggests that a big effort has been carried out over time to determine this property of sediments, in particular when exploring surficial sediments. However, in many cases, the generated data is only partially and diffusively published,

with most of the information not being available to the scientific community or to stakeholders. Often, stakeholders spend extra resources to characterise areas (to be managed or under analysis) where basic background already exists but it is not easily available neither public. That is the case of a large number of sediment samples collected within the frame of diverse Portuguese, Spanish and European founded projects whose results for the Atlantic margin of the Iberian Peninsula (namely coast and adjacent continental shelf) remain unpublished.

Here, we present a recently created database, IBAM-Sed – Iberian Atlantic Margin Sediments Database. IBAM-Sed compiles all available information about the textural properties of surface sediments along the southern Atlantic coast and shelf of the Iberian Peninsula collected within the frame of 24 projects (Fig. 1). The samples include sediments from emerged beaches, estuaries, a coastal lagoon and the adjacent continental shelf. The objective of this compilation is to make all this information available to the community as it might support future decision making or even to serve as reference to assess the impact of

specific activities over time. To illustrate the latter, we show the potential use of some of the data included in the database with three examples. The first shows how the database can be used to asses changes in the bed-surface sediment distribution within the Guadiana river estuary after the construction upstream of a large-scale dam. The second example shows how the aforementioned database can be used to map and assess the resources within the adjacent shelf for sand mining for beach nourishment purposes. Finally, the third example shows how the database can also support the morphodynamic classification

of beaches using mean grain sizes.



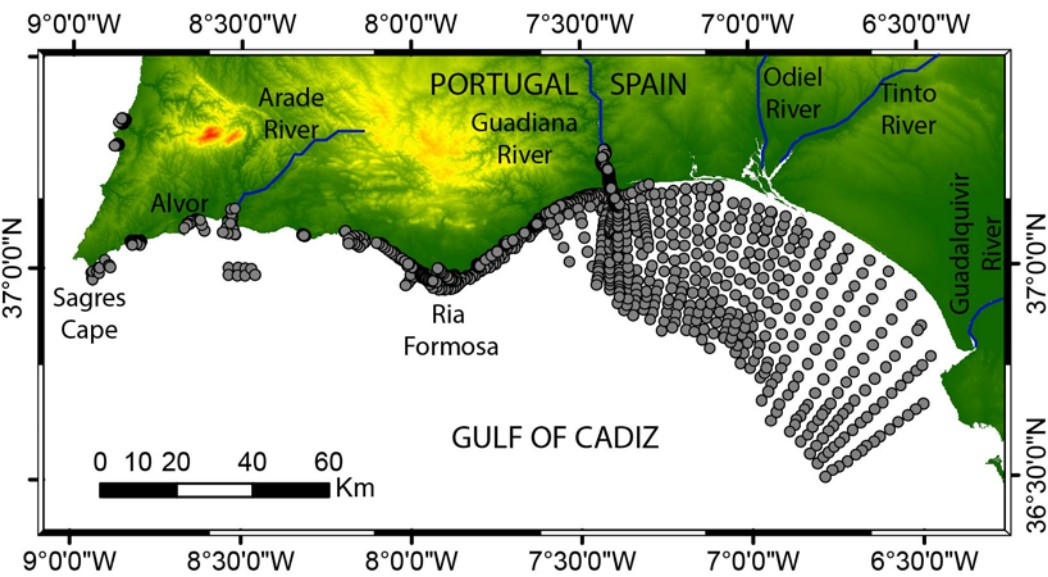

Figure 1: Distribution of the samples that conform IBAM-Sed database along the southern Atlantic coast of the Iberian Peninsula.

## 2 Data

### 2.1 Data origin

A total of 4727 samples were collected within the frame of 24 projects developed between 1996 and 2015 along the southern Atlantic coast of the Iberian Peninsula. The projects had different purposes, including fishing exploration, benthic characterisation, continental shelf evolution during the Holocene, coastal dynamics investigation or monitoring of nourished beaches. The variety of the projects explains the variety of environments that have been sampled and compiled within this database and also the non-uniformity of the sample distribution across the overall area of interest. Data collection was

performed by hand (at beaches and inlets) and using grab samplers (at the shelf). Table 1 summarizes the name of the projects, the sampled environments, the dates and the total number of samples collected within each research project.

### 2.2 Textural analysis

All collected samples were examined and analysed in order to determine their grain-size distribution. Due to the diversity of environments sampled and the subsequent variety of sediments, different methods were used to determine grain size

properties; i.e. sieving, pipette and sedimentation balance methods. Organic matter was removed from the samples by using hydrogen peroxide at increasing concentrations (10, 30, 80 and 130 vol/l). Fine (silt and clay) and coarse (sand and gravel) fractions were separated by wet sieving using a 63 µm (4 phi) sieve. For samples containing fine fractions (< 63µm) the



grain-size analysis was made using the pipette method that estimates the particle size distribution from the rate at which particles sink through a fluid. For those samples with coarser fractions (> 63 µm), the distribution of size fractions was obtained through dry sieving, by using a sieve column (at 0.5 phi or phi intervals) placed in a mechanical shaker for 20 minutes. The material retained in each sieve was weighted. In the specific case of EMERGE project, the main fractions were

separated by wet sieving and the sand fraction was analysed by using the sedimentation balance method.

The obtained and provided information on the grain size of the samples includes the main parameters used to texturally characterize the sediment samples, including the percentage per main size fraction (i.e. gravel, sand and mud), the mean grain size, the sorting, the kurtosis, the modes and the skewness. In most of the projects, the parameters were calculated applying

the Folk and Ward (1957) graphical method by using GRADISTAT (Blott and Pye, 2001), a programme that runs in Microsoft Excel, and is suitable for calculating particle size statistics for sieve or other methods frequently used for fine fractions. In other projects, with fine fractions analysed by the pipette method, the method of moments (Friedman and Sanders, 1978) was applied.

**2.3 Meta-information**

The database is divided into 24 events or files, each corresponds to the project within which framework the samples were collected. The information provided within the database includes references to the location of each sample (latitude, longitude and elevation or depth), the method used to collect the sample and to determine the grain size distribution, as well as the main textural parameters and the method used to calculate them, following the recommendations by PANGAEA. The data are available as txt files and openly accessible at https://doi.org/10.1594/PANGAEA.883104. Additionally, the data are also

available for visualisation at http://www.cima.ualg.pt/IBAM-Sed.

The diverse objectives of the projects behind the database explain the differences in the frequency of the retrieved samples. Samples related to projects focused on relatively stable zones or whose objectives do not include any monitoring strategy, have been collected only once (e.g. MESHATLANTIC project, Table 1). Alternatively, samples related to the monitoring of sandy beaches have been repeated over time following seasonal (e.g. CLIFF and BAYBEACH projects, Table 1) or even monthly

(e.g. MOSES project, Table 1) intervals during the duration of the project.

**3 Spatial coverage and grain size distribution**

The database covers the southern coast of Portugal and extends to the east until the continental shelf off the Guadalquivir river, covering therefore a large portion of the northern Gulf of Cadiz (Fig. 1). Therefore, the sampled area includes the continental shelf influenced by the main rivers in the Atlantic southern coast of the Iberian Peninsula, i.e. the Guadiana and the

Guadalquivir (Figure 1) and the Tinto and Odiel rivers. Towards the west, the shelf is more dispersedly sampled with samples off Ria Formosa and Arade estuary, off Alvor coastal lagoon and Sagres cape (Fig. 1). In addition, the database includes




surface sediment samples collected at the Arade and Guadiana estuaries and also inside the Ria Formosa coastal lagoon, and at several beach locations along the Algarve and Andalusian coast.

The overall data included in the database were never examined in a complete and integrated way. However, some of the samples were carefully studied and published as part of the results of the related projects. This is the case of the continental

5    shelf surface sediments off Guadiana-Guadalquivir rivers.

A surface sediment textural description has been elaborated in Gonzalez et al. (2004) using part of the data included in the database. Sandy deposits down to a depth of approximately 25 m (Fig. 2) characterize the continental shelf off Guadiana, despite several patches of sandy mud and mud (Gonzalez et al., 2004). The middle shelf is dominated by an extensive mud belt, consisting of very fine-grained clayey material. On the outer shelf, below 100 m, sediments are generally dominated by

10   sandy and silty clay. These are interrupted locally by large patches of sand and gravelly sand at the vicinity of the shelf edge.

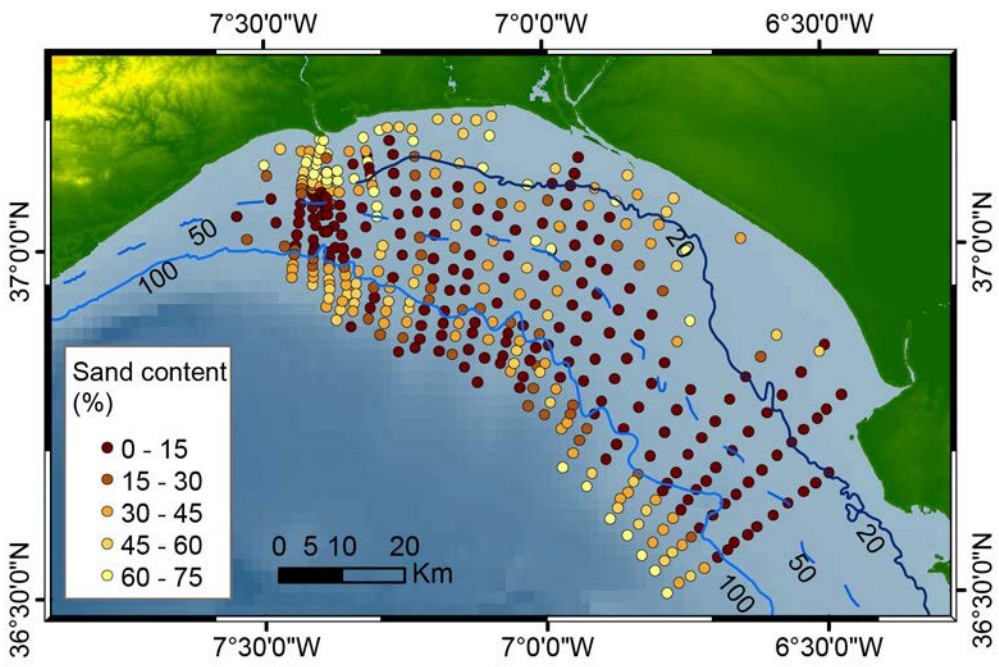

**Figure 2: Surface distribution of the content of sand within the area of the continental shelf more densely sampled within IBAM-Sed.**

Towards the west, the inner shelf surrounding the Ria Formosa barrier island system was systematically sampled (Fig. 3), allowing a detailed description of the surface sediments distribution (Rosa et al., 2013). Finest sediments (i.e. coarse silt; 31-63 µm) were found at the updrift margin of the Guadiana inlet while coarser sediments (i.e. very coarse sand; 1-2 mm) were found within the updrift end of the system at Olhos de Agua (Rosa et al., 2013). Between these two extremes, there is a general




increase in the mean grain size towards the east from fine sand dominating the western nearshore flank of the barrier island system to very coarse sand (Fig. 3). The latter trend appears related to a decrease in the incident wave energy and an increase of biogenic grains of the autochthonous benthonic communities (Rosa et al., 2013).

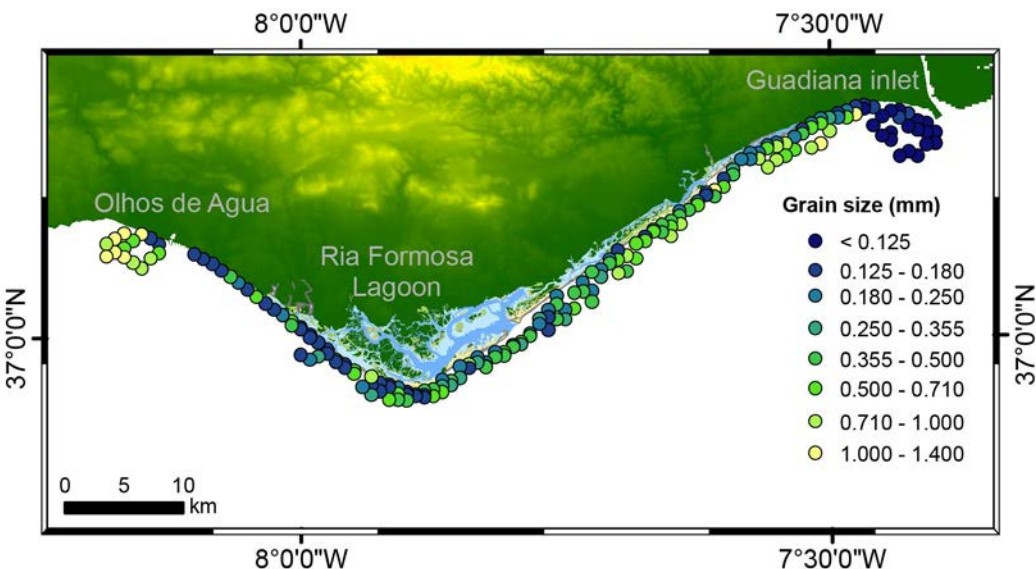

**Figure 3: Mean grain-size surface distribution on the inner shelf from Olhos de Agua to the Guadiana River mouth.**

The remaining areas of the southern Portuguese continental shelf have been poorly sampled with some samples collected off Arade Estuary, within the transition between the inner to the middle shelf, and few more samples off Sagres Cape (Fig. 1). The

10 results document coarser sediments, medium to very coarse sand, within the shelf off Arade Estuary, and medium to fine sands off Sagres Cape (Fig. 4).

Regarding the beaches along the southern Portuguese coast, a total of 24 beaches have been sampled and included within the database, covering most of the Algarve coast from the western margin to its eastern end (Fig. 4). Some of the beaches, in particular those within the peninsulas or barrier islands of Ria Formosa, have been re-grouped into 18 sites as shown in Figure

4. In some cases, the sedimentological characterization of most sampled beaches extended seaward, including the nearshore area (Table 1), and also landward, including the dune. In general terms, the mean grain-size of the subaerial beaches is dominated by medium sand (around 0.3 mm) along the western part of the studied area with a sudden increase at Galé-Olhos de Água region (Fig. 4). To the east of this region, the mean grain-size tend to decrease in the eastward direction (Fig. 4). Regarding the sorting of the analysed sediments, the results document an increase in the value of the sorting parameter that

translates in poorer selection towards the east with very well sorted (< 0.35) sediments within the western-most subaerial beaches and moderately well sorted (0.50-0.70) sediments in most of the rest of beaches (Fig. 4).



**Figure 4: Distribution of the mean grain-size at the different sampled beaches, Arade estuary and localized areas within the continental shelf. Mean-grain size and sediment sorting of the 24 sampled beaches (grouped into 18 sites) is also presented in the lower panels to more clearly show the longshore variability.**

As aforementioned, two estuaries and one lagoon are included in the database (Fig. 1). The Guadiana Estuary, from far the example better represented in terms of sampling density, is characterized by well-sorted medium sand at the lower estuary, while gravels can be found at the deepest locations, either mixed with sand or in small isolated pockets (Garel, 2017). Muddy





very fine sands are found within the transition between the deep channel and the shallow domains (Garel, 2017). The shallower marginal areas consist of muddy sediments. The few samples collected within the Arade Estuary point towards a high diversity of grain sizes; medium gravel grain sizes have been found within the inner estuary while silty to very fine sands can be found at the outer estuary (Fig. 4). The Ria Formosa lagoon is not completely represented as performed projects and thus related

5    sampling mainly focussed on monitoring tidal inlets and navigational channels (Fig. 1). The latter is mainly related to the fact that Ria Formosa hydrodynamics and morphodynamics are dominated by the multi-inlet system with six tidal inlets from which two have been artificially fixed, three have been artificially relocated while one evolves naturally. The results document the presence of heterogeneous sediments within the Ria Formosa tidal inlets and sampled channels ranging from very fine gravels to very fine sands (Fig. 5), reflecting the differences on hydrodynamics.

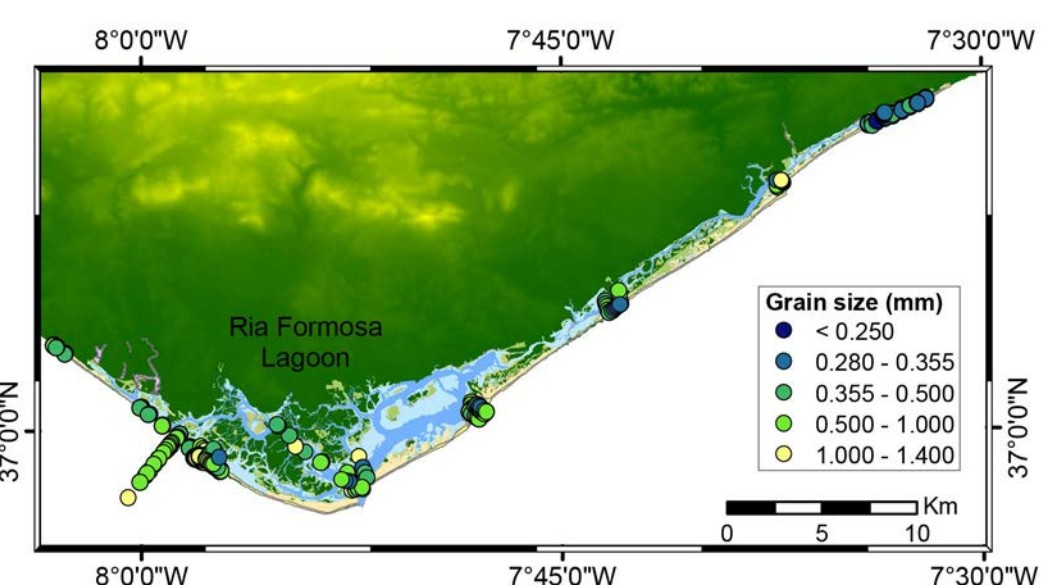

**Figure 5. Distribution of sediment samples and their mean grain-size within the Ria Formosa barrier island system.**

## 4 Application of the database

15    The present work also intends to illustrate through three examples the utility of the database here presented. The first case exemplifies how the database can be used to define a baseline situation to assess changes in the bed-surface sediment distribution. For that, we have used the results of the sediment textural analysis within the Guadiana river estuary collected before the construction upstream of a large-scale dam. The second case shows how the aforementioned database can be used to find the sand and gravel mining resources within the adjacent shelf for beach nourishment purposes. Finally, the third




example shows how the data from beaches can be used in combination with other sources of information, beach slope and incident wave characteristics in this case, to define the typical morphodynamic state of the sampled beaches.

## 4.1 Setting up baseline conditions

On February 8th 2002 the Alqueva dam was inaugurated in the Guadiana River with the aim to reinforce the capacity of hydroelectric production, to develop tourism, to promote regional employment, to organize intervention in environmental and patrimony domains, to fight physical desertification and climate change, to modify the agriculture model of south Portugal and to regularize river flow (Morais, 2008). With this new dam, built 60 km upstream from the estuary head, the regulation of the river flow increased from 75% to 81% (Rocha et al., 2002). The latter had significant consequences over the downstream phytoplankton communities (Domingues et al., 2014;Domingues et al., 2007), the volume of coarse sediment exported from upstream (Dias et al., 2004), and might have consequences on fish populations (Morais, 2008). The reduction of coarse sediment export to the estuary from upstream sources and the river flow regulation with the reduction (absence) of floods since 2001 could also provoke a change in the grain size distribution within the estuary itself that have not yet been assessed (Dias et al., 2004).

The samples were collected between September and October 2000 in the lower Guadiana estuary, together with a bathymetric survey (Fig. 6). The distribution of the main sediment components (i.e. mud, sand and gravel) shows a clear dominance of the sand fraction over the rest of the components clearly concentrated within the mid-channel (Fig. 6). The percentage of mud gradually increases upstream while the gravels appear to concentrate at the deepest areas of the estuary. This situation represents a baseline image of the conditions of the estuary prior to the construction of the Alqueva dam built within the Guadiana basin. The baseline situation presented here can be used to assess changes derived from the dam construction by comparing with a new grain size survey and analysis. That would provide insights on, for instance, areas of increasing/decreasing deposition of coarser (marine) sediments versus increasing/decreasing deposition of finer sediments due to the lower fluvial hydrodynamics.



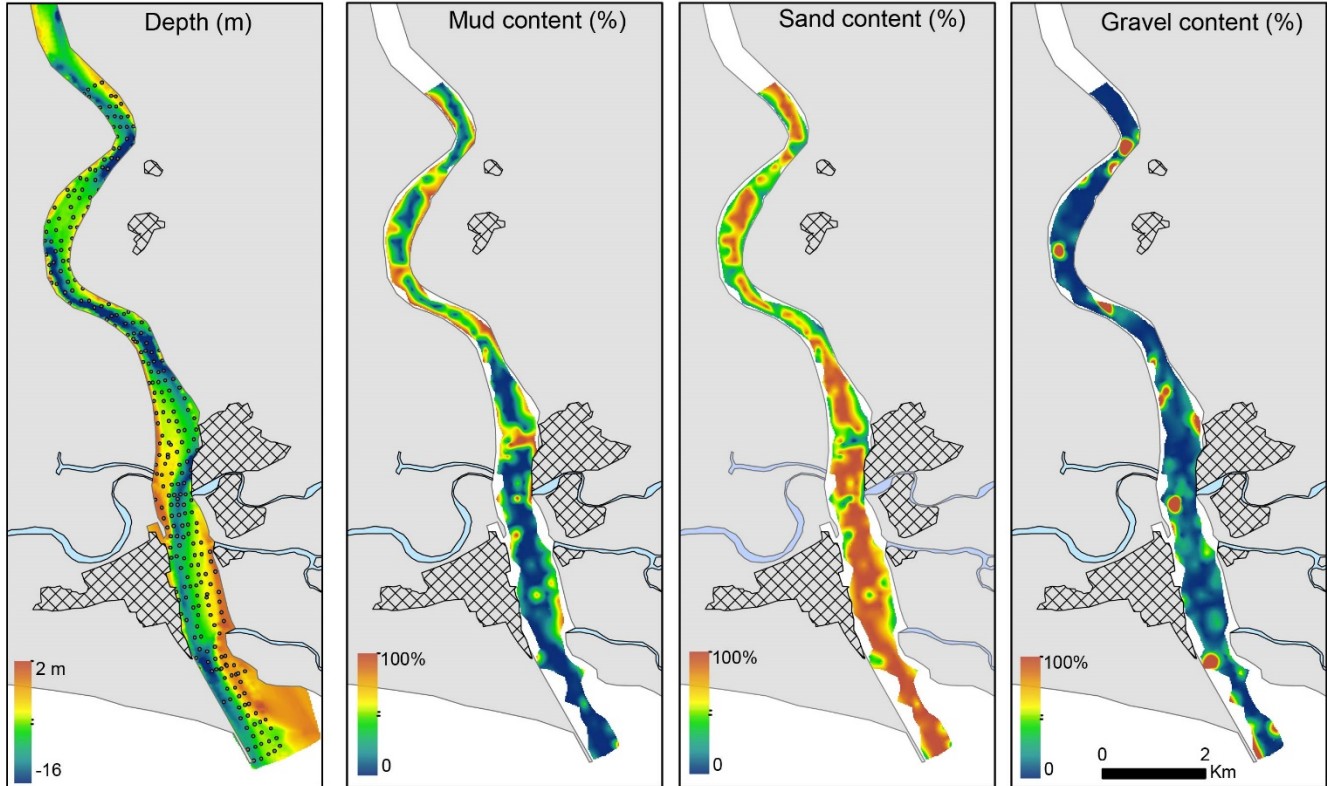

**Figure 6: Sediment distribution along the lower Guadiana estuary and contents of the main sediment fractions; mud, sand and gravels.**

## 4.2 Identification of sand banks

Beach nourishment is recognized as an environmentally friendly method of shore protection, being the method of choice for shore protection along many developed coasts with eroding beaches (Finkl and Walker, 2005). It is a practical method use for protecting against storm induced coastal flooding or erosion and structural shoreline retreat. It is also often used for shifting the shoreline seaward and for widening recreational beaches. Beach nourishment projects are complicated technical procedures that require careful preparation for successful execution of site-specific engineering design (Finkl and Walker, 2002). The grain size of the borrow material is one of the most important factors for optimizing beach nourishment, with several authors agreeing that coarser grain sizes produce steeper, more stable, and longer-lived fills (Finkl and Walker, 2005). Sand used at beach nourishment projects is mostly dredged from borrow areas at the seafloor (nearshore and inner shelf). That sediment should be obtained at depths greater than the local depth of closure (depth bellow which no relevant sediment exchanges occur between the nearshore and the adjacent shelf). Other sources of sand include inlet dredging, channel dredging or sand retained



at cross-shore obstacles (e.g. harbour jetties or breakwaters). The Portuguese law defines that all sand (with acceptable or good quality) extracted from the coastal area and off to one mile from the coast must be used as beach nourishment.

To assure that beach nourishment projects have a higher possibility of success it is thus necessary to have consistent and detailed data on both beach and borrow areas grain size and distribution. The presented database includes both and can then
serve as a tool to plan beach nourishment by providing information on the grain sizes and their variability from beach to beach and by allowing an easy comparative analysis of the compatible sediments located at the shelf or even at channels and inlets, which can be subjected to potential dredging. Figure 7 shows two areas of compatible sediments to exemplify the utility of the database: 1. a beach located downdrift Guadiana river mouth and whose retreat forced the authorities to nourish in 2005 (Garel et al., 2014), and 2. an area located at the shelf, seaward of the depth of closure, which in this area is located circa 10 m water
depth (Dolbeth et al., 2007), where sediments have compatible sizes and very low fine contents. It can also be observed that to the east the sediments start to have a larger percentage of fine material turning them unsuitable for the nourishment purpose.

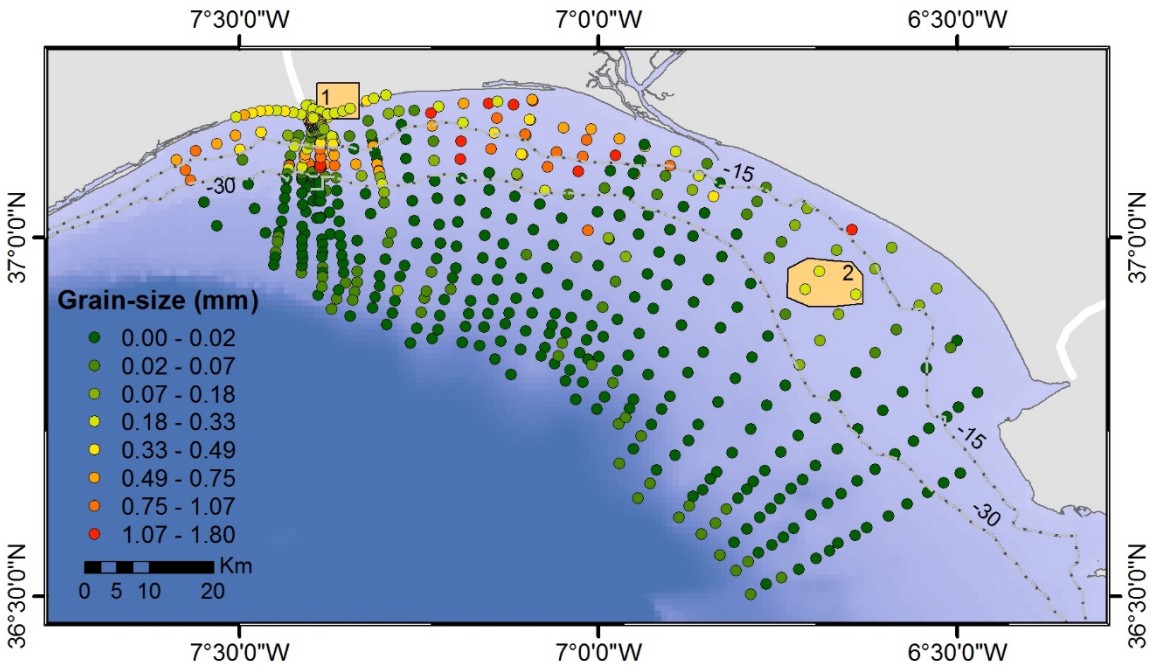

**Figure 7: Surface mean grain-size within the sampled continental shelf and adjacent beaches with evident problems of land loss. The**
**orange boxes show (2) an area within the continental shelf whose sediments are compatible with the sediments at the beach presenting**
**erosion problems (1).**

## 4.3 Characterizing beach morphodynamics

Beach morphodynamics refers to the mutual adjustment of hydrodynamic processes (principally shoaling-breaking swash
waves, tide, and wind), the sediment of the beach environment, and any other boundary conditions, in an attempt to maintain





a dynamic equilibrium (Wright and Thom, 1977). The development of a morphodynamic approach to beach studies in the 1970s provided a major new paradigm that transformed the way beaches were studied and allowed a considerable progress on the understanding and study of beach systems (Short, 1999). This notion has resulted in the formulation of beach type classifications that recognise and categorize the different morphodynamic signatures within distinct beach morphologies or

states, and link these to parameterisations related to key environmental conditions, namely wave climate, tidal regime and beach sediment characteristics. The most widely used of these models is the so-called Australian beach classification model, originally formulated by Chappell and Eliot (1979), Short (1978) and Wright et al. (1979), and subsequently modified by Wright and Short (1984). This approach enabled the full spectrum of wave-dominated micro-tidal beach systems and types to be identified and characterized and is utilized to examine beach response at different time scales. Yet, the later did not include

beach signatures derived from the effect of tides until Masselink and Short (1993) introduced a parameterisation of the relative importance of tides and waves by the relative tide range (RTR) given by *MSR/$H_b$*, where MSR is the mean spring tide range and $H_b$ is the modal breaking wave height. Therefore, this model classifies beaches using two dimensionless parameters: the RTR and the dimensionless fall velocity $\Omega = H_b/w_sT$, where $w_s$ is the sediment fall velocity according to Ahrens (2000) and T is the wave period.

Figure 8 shows the location of the 18 groups of sediment types derived from the 24 sampled beaches whose sediments were characterized using the mean grain-size from the present database and the result of the morphodynamic classification of those beaches after applying the required additional parameters (i.e. $H_b$, T, MSR). Wave climate characterization for the beaches at the south coast (beaches 4 to 18 in Fig. 8) was based on the available dataset from the Faro buoy for the period 1993-2013, located offshore Ria Formosa at 93 m depth (Fig. 8), for the beaches located along the south oriented beaches. The wave

climate of the beaches located along the western coast (beaches 1 to 3 in Fig. 8) was characterized using offshore hindcast wave data covering the same time interval (SIMAR dataset, code 5006023) provided by Puertos del Estado were used to characterize wave climate along the western coast. Wave refraction and shoaling were calculated using linear wave theory and combined with the breaking criterion $Y_b = H_b/d_b = 0.78$ to estimate $H_b$ at each beach considering beach orientation, where $d_b$ is the breaking depth. Waves approaching a particular beach with an angle greater than 70 degrees were not considered for

the analysis as the objective was to find the modal wave dominance at each beach. The classification documents a longshore variability in the type of beaches suggesting the Low Tide Terrace and the Low Tide Bar types as the more frequents within the southern coast. Yet, the classification clearly separates the beaches at the eastern flank of Ria Formosa, and in the vicinity of Sagres (Low Tide Bar, mostly intermediate) from the rest of beaches within the southern coast (mostly reflective), which are included within the Low Tide Terrace type (Fig. 8). This separation appears mostly related to the different nature of the

sediment or grain-size, which is greater within the second group (Fig. 8). In turn, the beaches located along the western coast, have been classified as Barred beaches, showing a tendency to more dissipative morphotypes. In this case, the greater wave break height within this area seems to mostly explain this result.



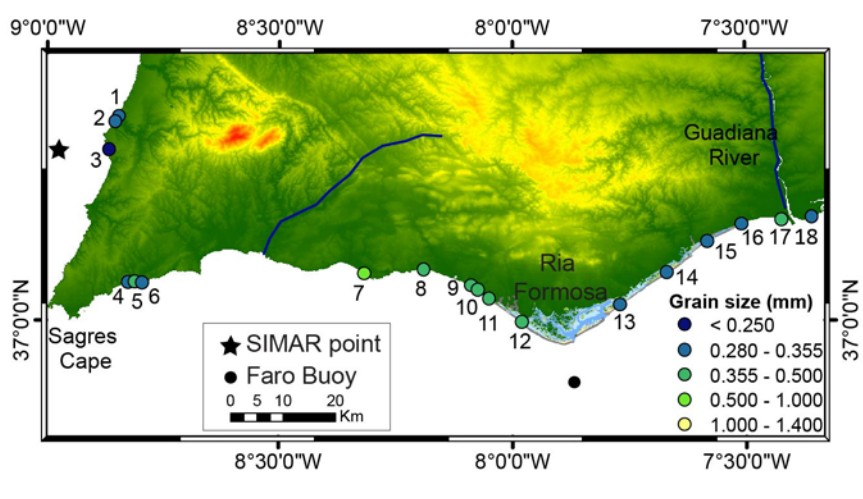

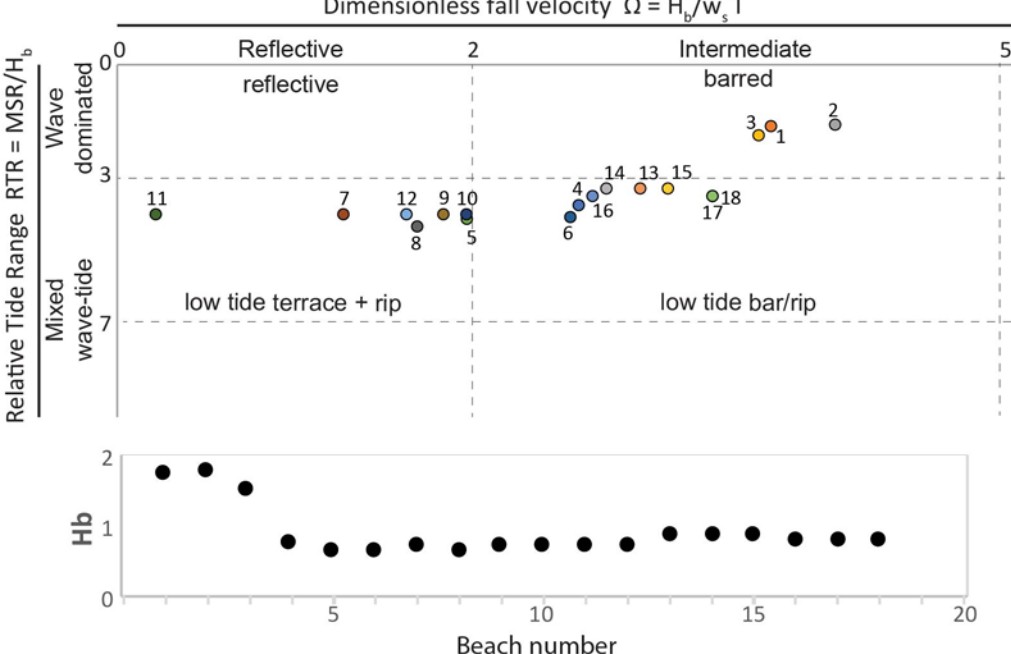

**Figure 8: Map showing the location and mean grain-size of the sampled beaches and the result of the morphodynamic classification using the Masselink and Short (1993) model. Finally, the lower graph shows the longshore variability of the modal breaking wave height used for the classification.**

## 5 Conclusions

A new database, IBAM-Sed, including the textural characterization of sediment samples along the southern Atlantic coast of the Iberian Peninsula is presented. The database compiles the results from 24 projects within the frame of which a total of 4727



samples have been collected between 1996 and 2015. The compiled database includes samples from diverse sedimentary environments from emerged coastal barriers (including dunes) to the outer continental shelf, integrating sediments as well from estuaries, and a coastal lagoon.

The utility of the compiled database is demonstrated through three examples that show how this type of information can be
used to characterize baseline conditions for further comparison analysis to assess the impact of anthropogenic activities within a basin over the surface distribution of sediments or even the impact from climate related variability. Alternatively, the database is shown to contribute to decision making by providing the needed information during the selection of potential areas for dredging for nourishment purposes. Finally, it is shown how the database owns relevant information to apply morphodynamic models of beach classification as sediment texture states as one of the most important parameters on the determination of
morphodynamic beach types.

## 6 Acknowledgments

The authors would like to thank all people involved in the collection of the samples integrating the present database. We also acknowledge to Puertos del Estado (Ministerio de Fomento, Spain) for providing the access to the SIMAR database. Susana Costas is founded through the "FCT Investigator" program (ref. IF/01047/2014).

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

**Table 1: Summary of the projects contributing information to the present database of surficial sediments, including the type of sampled environment, dates, number of samples and the technical DOI reference for each particular data set within the entire database. All data sets are aggregated in a parent set, which represents the entire dataset and whose citation DOI is doi.pangaea.de/10.1594/PANGAEA.883104**

| Project | Sampled Environment | Date(s) | Nº Samples | Data set technical DOI |
|---|---|---|---|---|
| BAYBEACH | Beach | 2007 to 2012 | 403 | https://doi.pangaea.de/10.1594/PANGAEA.883092 |
| MESHATLANTIC | Continental Shelf | 2011 | 22 | https://doi.pangaea.de/10.1594/PANGAEA.883091 |
| SIMCO | Estuary | 2015 | 22 | https://doi.pangaea.de/10.1594/PANGAEA.883079 |
| GESTEPESCA | Beach (nearshore) | 2005 | 212 | https://doi.pangaea.de/10.1594/PANGAEA.883073 |
| CLIFF | Beach | 2010 to 2012 | 218 | https://doi.pangaea.de/10.1594/PANGAEA.883083 |
| EROS | Beach | 2011 to 2012 | 247 | https://doi.pangaea.de/10.1594/PANGAEA.883090 |
| MOSES | Beach | 2011 | 1167 | https://doi.pangaea.de/10.1594/PANGAEA.883084 |
| COASTMONITOR | Beach | 1996 to 1998 | 455 | https://doi.pangaea.de/10.1594/PANGAEA.883098 |
| REQLAGOONRF | Beach | 1999 to 2000 | 100 | https://doi.pangaea.de/10.1594/PANGAEA.883112 |
| NEWSLUISINLET | Beach | 1997 | 45 | https://doi.pangaea.de/10.1594/PANGAEA.883080 |
| INDIA | Beach, Lagoon | 1997 to 1999 | 260 | https://doi.pangaea.de/10.1594/PANGAEA.883076 |
| CROP | Lagoon, Beach, Inner Shelf | 2001 to 2004 | 285 | https://doi.pangaea.de/10.1594/PANGAEA.883075 |
| BERNA | Lagoon | 2005 to 2008 | 20 | https://doi.pangaea.de/10.1594/PANGAEA.883093 |
| MICORE | Beach | 2009 | 14 | https://doi.pangaea.de/10.1594/PANGAEA.883081 |



| RUSH | Beach, Lagoon | 2012 to 2013 | 15 | https://doi.pangaea.de/10.1594/PANGAEA.883078 |
|---|---|---|---|---|
| GRADBENTHICRF | Lagoon | 2015 | 9 | https://doi.pangaea.de/10.1594/PANGAEA.883077 |
| IDEM | Lagoon | 2006 to 2007 | 66 | https://doi.pangaea.de/10.1594/PANGAEA.883094 |
| INLET-IPTM | Lagoon | 2001 to 2005 | 221 | https://doi.pangaea.de/10.1594/PANGAEA.883082 |
| SHORE | Beach | 2013 to 2014 | 65 | https://doi.pangaea.de/10.1594/PANGAEA.883103 |
| SIRIA | Continental Shelf | 1999 to 2001 | 522 | https://doi.pangaea.de/10.1594/PANGAEA.883074 |
| EMERGE | Estuary | 2000 | 284 | https://doi.pangaea.de/10.1594/PANGAEA.883087 |
| SYMPATICO | Estuary | 2008 | 6 | https://doi.pangaea.de/10.1594/PANGAEA.883088 |
| CRIDA | Beach | 2003 | 48 | https://doi.pangaea.de/10.1594/PANGAEA.883086 |
| CIRCO | Continental Shelf | 2010 | 21 | https://doi.pangaea.de/10.1594/PANGAEA.883085 |