# Peer review of "Surficial sediment texture database for SW Iberia Atlantic Margin"

_Earth System Science Data, 2018_

## Referee Comment (RC1) · Anonymous Referee #1 · 6 Apr 2018

The paper entitled "Surficial sediment texture database for SW Iberia Atlantic Margin" represents an important initiative from the researchers of the University of Algarve in disseminating scientific information to the community. The data are freely available under the Pangaea system. Some minor comments on the data that are present in the paper: the authors only present mean grain size information (obtained either by Folk & Ward or Moments methods). Are these values, obtained by different techniques, merged in the maps? Maybe this information could be available in the figure captions. Also, I would like to read about the grain size distributions, with information about modality and dispersion of the grain size around the mean (such as standard deviation values). I recommend the publication after these minor corrections

2018.

---

## Author Comment (AC1) · 15 Apr 2018

**Reply RC1 queries**

RC1: The authors only present mean grain size information (obtained either by Folk & Ward or Moments methods). Are these values, obtained by different techniques, merged in the maps? Maybe this information could be available in the figure captions.

Authors: We understand the problem raised by Reviewer 1 as the presented figures show the mean grain size of the sediments collected within different projects, and indeed in one case (i.e. Figure 7) it shows the results after Folk and Ward and the moments method.

Changes: We will add a reference to the methods used to calculate the textural parameters of the samples displayed within each figure, and for the particular case of Figure 7, we will specify which have been estimated by applying the moments and Folk and ward method.

RC1: I would like to read about the grain size distributions, with information about modality and dispersion of the grain size around the mean (such as standard deviation values).

Authors: The query raised by the reviewer is pertinent as in many cases the information about the mean grain size cannot help to understand how well sorted or not are the sediments, which in turn might have important implications from the hydrodynamic or sediment source points of view, for example.

Changes: In order to meet this query, we will include the description of the distribution of grain size over the populations described within the text, and we will also add a new panel to figures 3, 4, 5, and 7 that will represent the sediment sorting, as indicator of the grain size distribution of the sediments.

---

## Short Comment (SC1) · 24 Apr 2018

[revised manuscript text omitted]

---

## Referee Comment (RC2) · Anonymous Referee #2 · 3 May 2018

The paper entitled "SurïnËĞAËŻcial sediment texture database for SW Iberia Atlantic Margin" by Costas et al. is an important contribution to the scientific and technical publics, making available grain size datasets that can be user beyond the examples mentioned by the authors. In this context I strongly recommend its publication. However, some corrections/clarifications (technical and scientific) must be done, addressed in the several comments I've made in the attached text.

---

## Editor Comment (EC1) · H Grobe (Editor) · 3 May 2018

Dear Referee #2, thank you very much for reviewing the submission of Costas et al. to ESSD.

In your interactive comment you refer to "However, some corrections/clarifications (technical and scientific) must be done, addressed in the several comments I've made in the attached text."

Sorry to say, that author as well as editors can not find/access your attached text. Please clarify.

With kind regards Hannes Grobe (topical editor)

---

## Short Comment (SC2) · 7 May 2018

Dear Hannes The text is in SC1. Regards Maria

---

## Author Comment (AC2) · 8 May 2018

**Reply to SC1/RC2 comments**

**Reviewer**: Line 17 page 3 – grain size distribution performed by pipette only in the fine fraction?

Answer – we understand the question of the Reviewer, as the way this part of the text was written it is not clear enough. The pipette was used only for those fractions below 63 microns. We will review the text accordingly.

**Reviewer**: Line 2 page 4 – not clear if this refers to samples with only coarser fractions

Answer – this is linked to the above comment as stated by the Reviewer. We will improve the writing of this part accordingly to explain that coarser fractions, from those samples with a relatively high % of fine sediments.

**Reviewer**: Line 3 page 4 – grain size analysis at phi or half phi, do the authors state this within the metadata?

Answer – we will add a comment to the database with the information about the projects whose textural parameters have been obtained each half (i.e. CRIDA HM, EMERGE, SIRIA, SIMPATICO) or one phi. This info will be added as well in the manuscript.

**Reviewer**: Line 5 page 4 – the results of the analysis were merged together to obtain the textural parameters?

Answer – the results from each grain fraction, even obtained from different methods (e.g. pipette, dry sieving) were merged to obtain the textural parameters. The latter was done using Gradistat or applying the equations. This will be better explained in the text.

**Reviewer**: Line 11 page 4 – it is not clear the writing, you say that sieving is a method used for fine fractions? Clarify what other methods you refer to

Answer – we will re-write this part of the text as it seems confused as raised by the Reviewer.

**Reviewer**: Line 12 page 4 – also with Gradistat?

Answer – no, in this case that tool was not applied, but the original equations were applied to the obtained data, following the arithmetic method of moments).

**Reviewer**: Line 30 page 4 – correct the refce to the figure to "Fig. 1"

Answer – we will fix this.

**Reviewer**: Line 10 page 5 – the authors call sand to material that has a max of 75% sand (Fig. 2)

Answer – here, the authors are just citing from a previous work, where the grain size distribution within this area had been carefully discussed. We have now gone back to the original manuscript and changed the description as our image and the image from this manuscript documents. In fact, the authors refer to sand but indeed these are represented by patches of *muddy sand* and *slightly muddy sandy gravel*.

**Reviewer**: Line 19 page 5 – fix the name of the site to "Água"

Answer – we will fix this.

**Reviewer**: Line 6 page 6 – why do the authors use mm in the manuscript instead of phi units as in the database?

Answer – we have decided to refer to mm within the manuscript as we considered it more intuitive. We only describe from a very general perspective the database, so we tried to make it as simple as possible. To fix this mis-understanding, we will add a clarification note to the manuscript explaining that the description will be done in mm.

**Reviewer**: Line 17 page 6 – the beaches are dominated by medium sand and not the mean grain-size of the beaches…

Answer – we will fix this.

**Reviewer**: Line 17 page 6 – a sudden increase of what?

Answer – We will add more information to make sure that the readers will not get lost.

**Reviewer**: Line 18 page 6 – to which values does the trend decrease? And the peak at 12?

Answer – we will give more details about this spatial distribution as asked by the Reviewer, indicating the magnitude of grain size reduction and the existence of a peak at site 12.

**Reviewer**: Line 20 page 6 – the sorting is in phi units. The authors should refer the units they use all over the text and dataset

Answer – as asked by the editor, we had removed the units from the sorting, skewness and kurtosis parameters in the database, the sorting has the same units as the mean, and by suggestion of the editor we left this field as +/-. We will add a reference to the units of the sorting within the manuscript to fix this concern timely raised by the Reviewer, also in figure 4.

**Reviewer**: Line 3 page 8 – aren't the coarse samples from Arade estuary not represented in figure 4? The legend ends at 1.4 and the authors refer here to medium gravels (4 to 8 mm).

Line 8 page 8 – within the tidal inlets, you refer to very fine gravels, which sizes? The same problem with the legend that ends in 1.4 mm.

Answer – These two problems with the legends identified by the Reviewer will be fixed.

**Reviewer**: Captions should have ":" instead of "."

Answer – this will be fixed.

**Reviewer**: Line 11 page 9 – change to exported

Answer – this will be fixed.

**Reviewer**: Line 21 page 9 – the coarser sediments, are all marine? Even in the inner estuary?

Answer – we understand the question of the Reviewer, we were just describing very generally the most frequent situation, as described in Garel et al (2009, ECSS): "Similarly to the situation at the Guadiana estuary, seaward-flowing currents dominate throughout the estuaries, in relation with the narrow and confined channel morphology, during periods of significant freshwater discharge (spring freshets). This mechanism supplies coarse grained sediment to the system from the upstream river and governs net seaward sediment transport within the estuary (Fenster and FitzGerald, 1996). Sand is exported to the nearshore at a yearly to centennial scale (Fitzgerald et al., 2000; Fenster et al., 2001; Brothers et al., 2008). This pattern of seaward sediment transport within rock-bound estuaries

does not fit into the conceptual models of wave- and tide-dominated systems (Fitzgerald et al., 2000; Fenster et al., 2001; Brothers et al., 2008). The latter models consider estuaries as sediment traps (by contrast to river deltas) with landward sediment transport from the marine environment into the estuary (Boyd et al., 1992; Dalrymple et al., 1992).

However, and to avoid further misunderstanding we will not refer to the origin of the sediment within the manuscript; the reference to marine sediments will be deleted.

**Reviewer**: Caption Figure 6, change the ";" to ":"

Answer – this will be fixed.

**Reviewer**: Line 14 page 10 – change to "closure depth"

Answer – this will be fixed.

**Reviewer**: Line 19 to 22 page 12 – a sentence is repeated

Answer – this will be fixed.

**Reviewer**: Line 30 page 12 – change "which is" to "the later being"

Answer – this will be fixed.